# Anticapsular and Antifungal Activity of α-Cyperone

**DOI:** 10.3390/antibiotics10010051

**Published:** 2021-01-06

**Authors:** Connor Horn, Govindsamy Vediyappan

**Affiliations:** Division of Biology, Kansas State University, Manhattan, KS 66506, USA; cmhorn@ksu.edu

**Keywords:** α-Cyperone, essential oil, *Cyperus rotundus*, Candida spp., *Candida krusei*, *Candida auris*, *Cryptococcus neoformans*, anti-capsule, fluconazole synergy

## Abstract

Fungal infections affect 300 million people and cause 1.5 million deaths globally per year. With the number of immunosuppressed patients increasing steadily, there is an increasing number of patients infected with opportunistic fungal infections such as infections caused by the species of *Candida* and *Cryptococcus*. In fact, the drug-resistant *Can. krusei* and the emerging pan-antifungal resistant *Can. auris* pose a serious threat to human health as the existing limited antifungals are futile. To further complicate therapy, fungi produce capsules and spores that are resistant to most antifungal drugs/host defenses. Novel antifungal drugs are urgently needed to fill unmet medical needs. From screening a collection of medicinal plant sources for antifungal activity, we have identified an active fraction from the rhizome of *Cyperus rotundus*, the nut grass plant. The fraction contained α-Cyperone, an essential oil that showed fungicidal activity against different species of *Candida*. Interestingly, the minimal inhibitory concentration of α-Cyperone was reduced 8-fold when combined with a clinical antifungal drug, fluconazole, indicating its antifungal synergistic potential and could be useful for combination therapy. Furthermore, α-Cyperone affected the synthesis of the capsule in *Cryp. neoformans*, a causative agent of fungal meningitis in humans. Further work on mechanistic understanding of α-Cyperone against fungal virulence could help develop a novel antifungal agent for drug-resistant fungal pathogens.

## 1. Introduction

Fungal diseases contribute to significant morbidity and mortality in humans. It is estimated that nearly a billion people are infected with mucosal or superficial fungal diseases and 150 million people are affected with invasive fungal diseases [1]. The major invasive fungal infections include candidiasis, aspergillosis, cryptococcal meningitis, pneumocystis pneumonia, and histoplasmosis, which are responsible for over 1.6 million deaths globally [1]. People with compromised immune systems (e.g., HIV/AIDS, neutropenia) or with underlying health conditions (e.g., cancer, organ transplant, diabetes) are some of the high-risk groups affected most by invasive fungal infections.

Among the various fungal pathogens that affect humans, *Candida albicans* is the major fungus that is frequently associated with Candidiasis, causing a mortality rate up to 50% [2,3]. Humans are a natural reservoir for *C. albicans*, which colonizes on the mucosal surfaces (oral, GI, and urogenital tracts) and can disseminate systemically by their yeast-to-hypha conversion, leading to life-threatening invasive fungal infections. Other species of Candida (*C. glabrata*, *C. krusei*) also contribute significant infections that are intrinsically resistant to fluconazole [4]. An emerging Candida fungal pathogen, *C. auris*, is a major concern due to its pan-antifungal resistance, persistence in hospital environments as well as its invasive virulence potential with a high mortality rate (60%) [5,6].

A non-*Candida* fungal pathogen that also affects the immunocompromised is *Cryptococcus neoformans.* In the developing world, there are ~1 million cases of cryptococcal diseases per year, resulting in 675,000 deaths [7,8]. *Cryp. neoformans* is a major cause of fungal meningitis worldwide. *Cryp. neoformans* is further distinguished from *Candida* species by *Cryp. neoformans*’ ability to develop a capsule. The capsule acts as the main virulence factor of *Cryp. neoformans* by preventing the macrophages, dendritic cells, and neutrophils in the immune system from effectively containing the fungus [7,9,10,11]. In addition to its antiphagocytic functions, the capsule protects the fungus against desiccation in the environment [12]. *Cryp. neoformans* has also been reported to be able to develop increased resistance to common antifungals. A study in Uganda indicated that the MIC_90_ of fluconazole to *Cryp. neoformans* increased from 8 μg/mL to 32 μg/mL between 1998 and 2014 [13]. Furthermore, 31% of clinical isolates from that study were no longer susceptible to fluconazole [13]. While there are limited clinical antifungals available to manage these fungal pathogens, the increasing frequency of drug-resistance to available antifungals and their limitations including host toxicity, range of cellular targets, and oral bioavailability [3,14] warrants continued research and development for novel antifungal agents against these fungal pathogens.

During our screening of a focused library of plant compounds for antifungal activity [15] against *C. krusei* and other Candida species, we identified a fraction (#84) from *Cyperus rotundus* plant rhizome extract as a potential antifungal agent. *C. rotundus* (also known as nut grass or purple nut sedge in English) is found as a native species across Asia, Africa, and Europe [16]. *C. rotundus* is well known as a herbal remedy for various ailments in Roman, Chinese, and Indian traditional medicine [16]. Some of the many therapeutic characteristics attributed to essential oils (EO) extracted from *C. rotundus* are its uses as an astringent, diaphoretic, diuretic, analgesic, antispasmodic, anti-inflammatory, aromatic, stimulant, stomachic, vermifuge, tonic, and antimicrobial [16,17]. The EO extracted from *C. rotundus* rhizome contains various phytochemicals and α-Cyperone is the major component in it [16].

Various health benefiting properties of α-Cyperone have been described. According to Liu et al. [18], α-Cyperone showed a protective effect on lipopolysaccharide (LPS)-induced acute lung injury/inflammatory response in mice by downregulating the NF-κB and NLRP3 signaling pathways, mainly by upregulating SIRT1. In another study, the neuroprotective effect of α-Cyperone against H_2_O_2_-induced oxidative stress and apoptosis in dopaminergic neuronal SH-SY5Y cells was demonstrated [19]. Anticancer activity of α-Cyperone has also been reported [20].

Although various pharmacological activities of α-Cyperone against mammalian cells are known [18,19], the antifungal and anticapsular activities of α-Cyperone against fungal pathogens *Candida* and *Cryptococcus* species are unknown. In this study, we investigated the antifungal activity of α-Cyperone against various human fungal pathogens, its antifungal synergistic activity, and the anticapsular activity of α-Cyperone on *Cryp. neoformans*.

## 2. Results

### 2.1. Cyperus Rotundus Rhizome and α-Cyperone

To identify potent antifungal compounds, we used our in-house collection of medicinal-plant-derived compounds [15] to screen against Candida species and *Cryp. neoformans*. The compounds were dissolved in DMSO and used at different concentrations in 96-well microtiter plates. Using the Clinical and Laboratory Standards Institute (CLSI) M27-A3 method [21] and Roswell Park Memorial Institute (RPMI) 1640 medium at 37 °C incubation statically for 24 h, we identified a fraction (#84) derived from *Cyperus rotundus* rhizome as a potential antifungal source. An image of *C. rotundus* and the rhizomes are shown in Figure 1a. Since fraction #84 was derived from water distillation of *C. rotundus* rhizome and is known to contain EO, we focused on one of its major components.

The EO of *C. rotundus* contains various components of terpene derivatives. According to Sofia et al. [16], the EO of *C. rotundus* contains 27 components including α-Cyperone (26.15%), ß-selinene (17.99%), and Cyperene (15.73%) [16]. Similarly, the EO from the rhizome of another species of *Cyperus* contained Cyperone (47.6%), alpha-pinene (18.8%), 1,8-cineole (14.5%), and caryophyllene oxide (7.3%) as major principles [22]. The composition and percentage of the EO can vary depending upon the geographical location of the plant and its growing conditions. Since α-Cyperone is the major component of *C. rotundus* (26%) [16,17] and its antifungal property is unknown, we obtained α-Cyperone commercially (Adooq Bioscience, Irvine, CA, USA) and investigated it further. The α-Cyperone used in this study was 99.45% pure and an analytical high performance liquid chromatographic (HPLC) chromatogram is shown in Appendix A. The chemical structure of α-Cyperone is shown in Figure 1b. The minimal inhibitory concentration (MIC) of α-Cyperone against *C. albicans*, *C. krusei*, *C. glabrata*, and *Cryp. neoformans* was 500, 250, 500, and 500 µg/mL, respectively. Although these MIC concentrations for α-Cyperone were higher than the MICs of clinical antifungal drugs known, we wanted to investigate α-Cyperone further for its antifungal synergy and its effect on the capsule of *Cryp. neoformans*.

### 2.2. Antifungal Activity of α-Cyperone

We next determined the growth inhibitory effect of α-Cyperone against *C. krusei* using the Bioscreen-C growth monitor where the fungal growth rate was recorded at 30 min intervals. At least two different concentrations were used. Control (dimethyl sulfoxide, [DMSO] solvent only) wells were also included in parallel. Results shown in Figure 2a suggests that α-Cyperone inhibited the *C. krusei* completely at a 125 µg/mL concentration. The reduced inhibitory concentration may probably be due to the growth conditions involving mild shaking (10 s) of the plates in the Bioscreen-C instrument before registering the absorbance every 30 min. To determine if α-Cyperone is fungicidal or fungistatic, an aliquot of *C. krusei* and *Crypt. neoformans* cells from the MIC determination wells were serially diluted, and spot tested on agar medium. The results indicate that α-Cyperone was fungicidal (Figure 2b) for both fungi. There were no growth of fungi from α-Cyperone treated cells on yeast peptone dextrose (YPD) agar.

### 2.3. Synergistic Activity of α-Cyperone with Fluconazole

Since *C. krusei* and other species of Candida are resistant to fluconazole [4], antifungal compounds with potential synergistic activity will be useful in combination therapy. Antifungal synergy was determined using a broth microdilution based checkerboard assay [23]. Results presented in Table 1 indicate that α-Cyperone acts synergistically with fluconazole and reducing the MIC of both the antifungal agents from 31.3 µg/mL to 1.95 µg/mL for fluconazole (16-fold reduction) and from 250 µg/mL to 31.25 µg/mL for α-Cyperone (8-fold reduction). The FICI value was calculated to 0.31, which suggests synergy [23].

### 2.4. Effect of α-Cyperone on Fungal Membrane Integrity

Since α-Cyperone is an EO and the EOs affect mostly the cellular membrane (e.g., fungi) [24], we performed the propidium iodide (PI) uptake assay for α-Cyperone treated (125 µg/mL) *C. krusei* cells. Cells with intact membranes are impermeable to fluorescent dye, PI, unless the membrane integrity is lost due to α-Cyperone exposure. As a positive control for PI staining, heat-killed (boiled for 5 min) *C. krusei* cells were used. As expected, the fluorescence microscopy analysis of heat-killed cells showed uptake of PI dye. Unexpectedly, α-Cyperone treated cells did not show fluorescence, which suggests that α-Cyperone does not affect the cell membrane (Figure 3). *C. krusei* did not show any autofluorescence under similar conditions (not shown).

### 2.5. α-Cyperone Reduces the Thickness of the Cryptococcus neoformans Capsule

Unlike *Candida* species, *Cryptococcus* sp forms a capsule, a major virulence factor of this fungal meningitis pathogen. To visualize capsules better and to evaluate the effect of α-Cyperone without inhibiting the fungal growth, we used a low dose of α-Cyperone in capsule-inducing growth conditions. After India Ink exclusion staining, capsules were visualized under a microscope, and images were recorded. Results from this experiment revealed a significant reduction in capsule size in α-Cyperone treated samples (Figure 4a,b).

### 2.6. α-Cyperone Inhibits the Drug-Resistant Emerging Fungal Pathogen, Candida auris But Has No Effect against Bacteria

*C. auris* is an emerging drug-resistant fungal pathogen that causes both skin and invasive infections. Since α-Cyperone inhibits the growth of *C. krusei* (fluconazole-resistant), we wanted to determine if this EO could also affect *C. auris*. The Bioscreen-C growth monitoring system was used for this purpose. The results shown in Figure 5a suggest indeed that α-Cyperone prevented the growth of *C. auris* at the concentrations used. We next asked if α-Cyperone could affect the growth of Gram-positive or Gram-negative bacteria. We used paper discs containing α-Cyperone (60 µg/disc) on a lawn of fresh cells. We did not see the inhibition of *E. coli* or *S. aureus* bacterial growth around the discs (Figure 5b).

## 3. Discussion

Natural products and their scaffolds served as the primary source of many clinical drugs currently in use [25,26,27,28,29]. Notably, two of the three major clinical antifungals were from natural sources [27,30]. Natural products, therefore, offer great promise for discovering and developing future medicines. Natural sources, especially plants, are often exposed to pathogens or pests, and thus develop natural defenses against them by producing antimicrobial compounds. We have identified antifungal compounds earlier from medicinal plant-derived compounds and synthetic derivatives [15,31].

In this study, we report the EO α-Cyperone as an antifungal agent that inhibits the growth of various human fungal pathogens, albeit at a higher MIC. α-Cyperone is a major component in the *C. rotundus* plant rhizome [16,17]. Rhizomes are underground vegetative propagules that confront against various microbes/pathogens around them constantly in the soil and hence they are expected to contain several antimicrobial compounds. This plant rhizome (or tuber) contains hundreds of phytochemicals including several types of EO. While α-Cyperone is the major component and showed strong antifungal activity against human fungal pathogens, it did not show antibacterial activity against *E. coli* and *S. aureus*, at least under the conditions or doses tested in this study (Figure 2 and Figure 5). It is possible that the concentration used in the disc may not be sufficient to inhibit the bacterial growth. In general, the combination of different antimicrobial compounds exerts a stronger effect than their individual components. Some of the earlier studies conducted with total or crude extracts from *C. rotundus* showed antibacterial or antifungal activities ([16,17], references in these reviews) may affirm this notion.

Interestingly, α-Cyperone showed synergistic activity with one of the clinical antifungals, fluconazole (Table 1). The synergistic antimicrobial activity can result from the inhibition of two different targets or pathways. Fluconazole affects the synthesis of ergosterol in the fungal membrane. EO is also known to affect the cellular membrane by targeting the phospholipid [24]. Hence, we performed the PI uptake assay with the α-Cyperone treated *C. krusei* cells and we did not find the PI uptake (Figure 3), suggesting α-Cyperone may not be targeting the fungal membrane. It is worth mentioning that α-Cyperone inhibits the proliferation of cancer cells through the ROS-mediated signaling pathways [19]. Fungi as eukaryotic cells, α-Cyperone may function similarly or to related pathways in them, which remain to be determined.

One of the unexpected findings of this study is the inhibitory effect of α-Cyperone on the capsule of *Cryp. neoformans*. While the MIC of α-Cyperone against *Cryp. neoformans* is high, at a lower concentration (16 µg/mL), it inhibited the capsule size significantly (Figure 4). Furthermore, determination of the capsulated cells in the control and α-Cyperone treated samples showed that while 85% of the cells were capsulated in the former, only 21% were capsulated (all of them were with reduced capsule size) in the treated sample (data not shown). The capsule is a major virulence factor in *Cryp. neoformans* and capsule-deficient strains were avirulent [10,11,32,33]. A mutant strain (*pkr1*) of the cAMP pathway produces larger capsules than the wild type and was virulent in an animal model [33]. The *Cryp. neoformans* capsule wall contains two major polysaccharides, glucuronoxylomannan (GXM) and galactoxylomannan (GalXM), in addition to a smaller amount of mannoproteins [9,12]. It is possible that α-Cyperone may interfere with the synthesis of these polysaccharides or via the ROS-mediated signaling pathways. Interestingly, we observed that α-Cyperone treated *Cryp. neoformans* cells were broken more efficiently than the control cells by glass beads-mediated homogenization (Precellys 24) (Appendix A), which may indicate that α-Cyperone can affect the cell envelope probably by causing a defect either in their synthesis or in their cross-linking processes to the cell wall/capsule during maturation steps [34]. Future experiments including gene expression and biochemical studies will help us to understand the antifungal mechanism of α-Cyperone.

In conclusion, we identified α-Cyperone as a fungicidal antifungal agent that can synergize with the activity of a fluconazole antifungal drug that can be used in combination antifungal therapy. While α-Cyperone inhibits drug-resistant fungal pathogens, it appears to have no effect on bacteria. Furthermore, α-Cyperone contains a novel property of inhibiting capsule synthesis in *Cryp. neoformans*, the human fungal meningitis pathogen.

## 4. Materials and Methods

### 4.1. Plant Source, Rhizome Extract, and α-Cyperone

Our group used a focused library of medicinal plant-derived compounds earlier [15]. Briefly, this library contains about 600 compounds fractionated on conventional Si-columns. Selected medicinal plants were collected mostly from Tamil Nadu, Southern India and authenticated by an ethnobotanical expert. The library of compounds contain partially purified compounds from ethanolic and aqueous extracts (including conventional Soxhlet distilled) of rhizomes of the *C. rotundus* plant. A commercial source of α-Cyperone (Cat# A15413) was obtained from AdooQ Bioscience (Irvine, CA, USA) and diluted in DMSO when required. Fluconazole was obtained from Sigma (St. Louis, MO, USA) and dissolved in water. Aliquots were stored at −20 °C as recommended.

### 4.2. Antifungal Assay Methods

Fungal strains (*C. albicans*, *C. krusei*, *C. glabrata*, *C. auris*, and *Cryp. neoformans*) [31] were grown on YPD medium (1% yeast extract, 2% peptone, 2% glucose) before using them for antifungal screening or test assays. The cultures were adjusted to required cell densities (1 × 10^6^/mL) in buffered RPMI 1640 medium with 50 mM glucose according to the CLSI method for yeasts [21]. Antifungal compounds (α-Cyperone and fluconazole or amphotericin B) were tested using two-fold dilutions in 96-well microtiter plates. Each assay was performed in triplicate and repeated at least twice. Plates were then incubated overnight at 37 °C for 24 h statically, and then had their absorbance values recorded using a multi-well plate reader (Victor-3 multimode plate reader, Perkin Elmer). YPD agar medium was used for spot test assays.

To determine the effect of α-Cyperone on the growth rate of *C. krusei*, we used a real-time growth monitoring system (Bioscreen-C, Oy Growth Curves AB Ltd., Helsinki, Finland). The plates were incubated at 37 °C without shaking except for 10-s of shaking before reading absorbance at 600 nm at 30-min intervals. Data were plotted using the GraphPad program.

### 4.3. Checkerboard Assay

Antifungal synergy was tested on *C. krusei* using the checkerboard titration method. Fluconazole and α-Cyperone were diluted at different concentrations in buffered RPMI 1640 medium (Gibco, Grand Island, NY, USA) containing 50 mM glucose. Fluconazole concentrations between 0.24 µg/mL to 62.5 µg/mL, and α-Cyperone concentrations from 7.8 µg/mL to 500 µg/mL were used. Two-fold serial dilutions of fluconazole were performed going left to right on a 96-well plate with two-fold dilutions of α-Cyperone being performed going from the top to the bottom of the plate. The far-right column was used as an α-Cyperone control, while the bottom row was used as a fluconazole control. The control columns were then compared to the MIC of the two compounds being used together in order to calculate the FICI as described [23]. Synergism was defined as FICI < 0.5; additivity FICI between 0.5 and 1; indifference FICI between 1 and 2; and antagonism FICI > 2 [23]. Experiment was repeated three times.

### 4.4. Propidium Iodide (PI) Uptake Assay

The fluorescent dye PI uptake assay was performed to understand the antifungal mechanism(s) of α-Cyperone. Control and α-Cyperone treated *C. albicans* and *C. krusei* cells (16 h treated) were incubated with diluted PI solution in dark conditions for 30 min at room temperature. Cells were washed with PBS and viewed under a fluorescent microscope (Leica, DM 6 B with a camera attached) using Texas red filter. Untreated cells (autofluorescence) and heat-killed cells (positive control) were also included. Similar results for *C. albicans* and *C. krusei* were observed and results only for *C. krusei* are shown.

### 4.5. Determining the Effect of α-Cyperone on Cryptococcus neoformans Capsule Size

The effect of α-Cyperone on the *Cryp. neoformans* capsule was determined using a capsule inducing growth medium. It was achieved by incubating *Cryp. neoformans* cells in microtiter wells containing Dulbecco’s Modified Eagle’s Medium (DMEM) at 37 °C in a cell culture incubator (5% CO_2_) for two days [35]. Since the MIC concentration of α-Cyperone was fungicidal, a sub-MIC concentration of α-Cyperone (16 µg/mL) was used. Culture wells without α-Cyperone (+solvent) served as the control. Triplicate wells were used and the experiment was repeated at least twice. Capsules were seen using India ink staining under a microscope (60 objective × 10× ocular lenses) and images were recorded using a camera attached to the microscope. The sizes of the capsules and cells in both the control and α-Cyperone treated cultures were calculated in µm using the scale bar function for scale measurements in the ImageJ software as described [35]. Statistical analysis of the data was performed using a two-tailed t-test with Welch’s correction (GraphPad). *p*-value < 0.0001 was considered statistically significant.

## Figures and Tables

**Figure 1 antibiotics-10-00051-f001:**
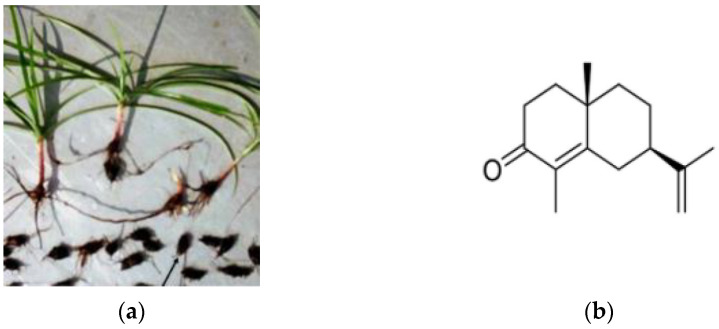
*Cyperus rotundus* plants and rhizomes (arrow) (**a**) collected in Southern India and the chemical structure of α-Cyperone (**b**) (from Peerzada et al. 2015; Huang et al. 2020). The molecular weight of α-Cyperone is 218.33 Da.

**Figure 2 antibiotics-10-00051-f002:**
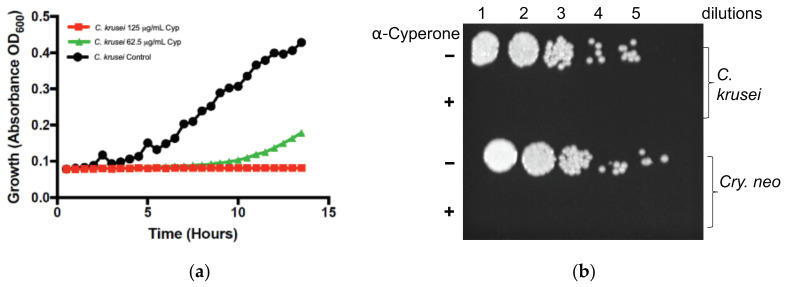
Antifungal activity of α-Cyperone against *Candida* and *Cryptococcus* species. (**a**) *C. krusei* was grown in honeycomb microtiter wells containing RPMI 1640 medium in the presence (62.5 µg/mL & 125 µg/mL) and absence of α-Cyperone for about 14 h at 37 °C. Fungus growth was measured by absorbance at optical density (OD) 600 nm using Bioscreen-C growth monitor. (**b**) Control (−) and α-Cyperone exposed (+) (MIC dose) fungal cells from microtiter wells were serially diluted and spot tested on YPD agar plates. Plates were incubated at 30 °C for 2-days and photographed. A representative image is shown.

**Figure 3 antibiotics-10-00051-f003:**
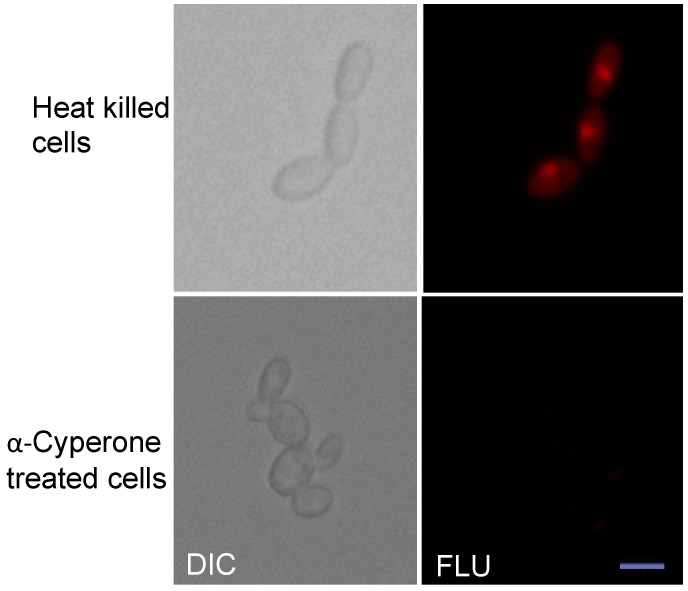
The membrane integrity of *C. krusei* fungus was unaffected by α-Cyperone. Positive control (heat-killed cells) and α-Cyperone treated (125 µg/mL) cells were incubated with propidium iodide (PI) fluorescent dye solution as described by the manufacturer (Invitrogen, Carlsbad, CA, USA). Cells were viewed under a fluorescence microscope (Leica DM 6 B) with an appropriate filter. Fluorescent (FLU) images and corresponding differential interference contrast (DIC) images are shown. Scale bar, 10 µm.

**Figure 4 antibiotics-10-00051-f004:**
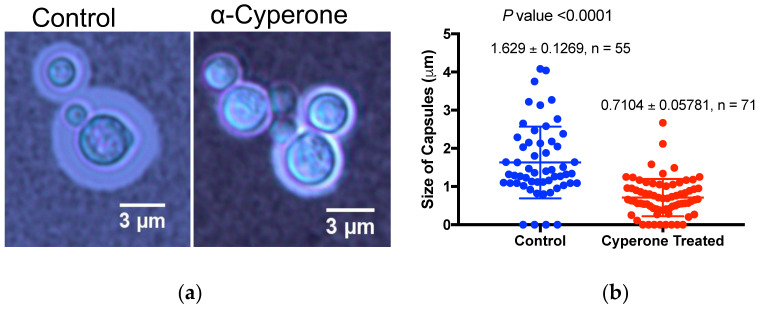
α-Cyperone inhibits the capsules in *Cryp. neoformans*. (**a**) Representative images of control (+ solvent) and α-Cyperone (16 µg/mL, sub MIC) exposed *Cryp. neoformans* grown in Dulbecco’s Modified Eagle’s Medium (DMEM) for 48 h at 37 °C. Capsules were visualized by India ink staining and examined under a microscope (Leica). (**b**) The size of capsules (in µm diameter) in control and α-Cyperone treated samples were determined using ImageJ software. Statistical analysis of the data was performed using a two-tailed *t*-test with Welch’s correction (GraphPad). *p*-value < 0.0001.

**Figure 5 antibiotics-10-00051-f005:**
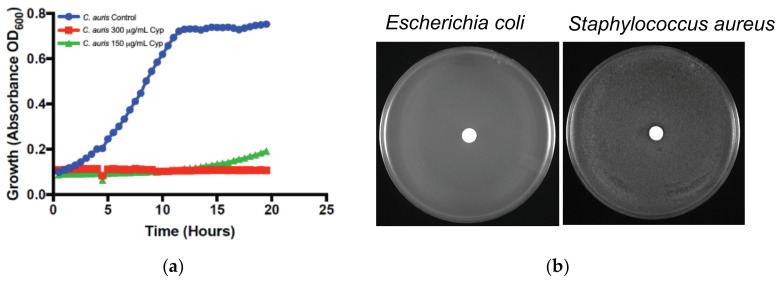
α-Cyperone affects drug-resistant *C. auris*, but has no effect on bacteria. (**a**) Inhibition of *C. auris* growth as determined by a growth monitoring system (Bioscreen-C). (**b**) Absence of antibacterial activity of α-Cyperone against *E. coli* and *S. aureus.* Sterile discs containing α-Cyperone (60 µg/disc) was placed immediately after spreading the bacterial cell suspension (100 µL, 10^9^/mL) on tryptic soy agar medium. Plates were incubated at 37 °C for 24 h and photographed. Lack of inhibitory zones around the discs was observed.

**Table 1 antibiotics-10-00051-t001:** Synergistic antifungal activity of α-Cyperone with fluconazole against *C. krusei*.

Antifungal	MIC by Checkerboard Assay (µg/mL)
α-Cyperone *	31.25
Fluconazole **	1.95
FICI ***	0.31

* The MIC of α-Cyperone against *C. krusei* was 250 µg/mL by the CLSI method; ** The MIC of *C. krusei* was 31.30 µg/mL; *** Fractional Inhibitory Concentration Index (FICI) determined by checkerboard assay.

## Data Availability

Not applicable.

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
