# Peer review of "Anticapsular and Antifungal Activity of α-Cyperone"

_antibiotics, 2021, doi:10.3390/antibiotics10010051_

Round 1

Reviewer 1 Report

The manuscript is with academic value by a focus of drug-resistant fungi inhibition. Important and necessary literature is properly cited to support the content flow. Meanwhile, certain technical aspects need to be improved.

  1. The manuscript viewed α-Cyperone as an essential oil, but it provided no information that validates this declaration. From the given methods to obtain the plant extract, we were told that α-Cyperone is a component of contain fractions from ethanolic and aqueous extracts of rhizomes of Cyperus rotundus plant. This is not consistent with the conventional essential oils unless other finger indices, like boiling point, of the compound were provided.
  2. Anticapsular analysis is a highlight of the investigation, indeed, but the scientific evidence supplied is not sound enough to validate this bioactivity disclosure. Capsule is the sporangium of various spore-producing fungi; its vigor can be explored via various phenotypic observations besides its size.
  3. The language of manuscript needs improvement since it seems not well edited.

Conclusion:

Major revision with necessary experiment supplements.

Author Response

Dear Reviewer 1:

We sincerely thank the reviewer for insightful comments and valuable time. We carefully read all the comments and addressed them in the revised manuscript. Additional details or methods were included in the texts which hopefully clarify the concerns. The revised texts were checked for grammatical errors. All the modifications were indicated with track changes in the attached word file.

We sincerely hope the revised version is improved and acceptable to the reviewer.

Reviewer 2 Report

In this concise paper, Horn and Vediyappan report their investigations on the antifungal activity of alpha-Cyperone, a plant-derived natural product. The topic is current, the scientific methodology is adequate and the findings should be interesting to the research community. However, the following major concerns should be addressed prior to publication.

1. In their original antifungal screen, the authors first identify a fraction of the extract of the Cyperus rotundus rhizome. They choose to proceed alpha-Cyperone to their more detailed investigation, specifying that it is the major component of the fraction (26%). What other compounds have constituted this fraction and in what percentages? Weren't there other compounds that could also be nominated for further investigation?

2. The discussion on the function of capsules (Cryptococcus) is somewhat limited. For example, it is not self-explanatory to me that a decrease of the capsule size would automatically mean a decreased virulence (intuitively, I would think that virulence is measured by capsule count, rather than capsule size). Please provide more literature context on this, to support your choice of the capsule size as an indicator of virulence. Also, it is unclear to me what the reader should see in Figure 4A: the two images are very similar (and blurry), the right one is slightly darker. Some reference scale/visual aid would be useful here.

3. In section 2.4, the authors establish that the inhibitory activity of alpha-Cyperone is not mediated by disrupting the integrity of the fungal membrane. What other mechanisms of action could come to mind? The work would be much more comprehensive if the authors could find this out and support their ideas by experimental results.

Minor

1. The authors should double-check the paper for typos (e.g. line 101: "may probably due to the growth") and incorrect references to figures/tables (e.g. line 105: "Figure 1b", line 148: "Figures a&b"). Also, for Figure 1b, I would suggest a larger font and thicker lines for the 2D images (I recommend the MarvinSketch program, which is free for academics, but there are many more alternatives).

Round 2

Reviewer 1 Report

All questions were well responded.

Reviewer 2 Report

The authors have revised and sufficiently improved the paper by providing more detail for certain parts of the discussion, correcting images, etc. I believe it is acceptable for publication in its current version. One minor error to correct during proofreading: line 92-93 says "the EO from [...] contained Cyperone [...] as major principles", I believe the correct wording here would be "major components" instead of "major principles".